# FairOCL: Fair Gradient Aggregation for Online Continual Learning

## Abstract

Online continual learning (OCL) aims to enable neural networks to learn sequentially from streaming data while mitigating catastrophic forgetting, a key challenge in which learning new tasks interferes with the retention of previously acquired knowledge. Although most existing approaches rely on memory buffers to replay past samples, training jointly on mixed data from different tasks often leads to gradient conflicts, which undermine model performance. To address this, we propose FairOCL, a framework that draws inspiration from fair resource allocation in communication networks. FairOCL formulates gradient aggregation across tasks as a constrained utility maximization problem and enforces fairness in the optimization process, allowing principled control over task prioritization. Extensive experiments on several standard benchmarks show that FairOCL achieves consistent improvements over state-of-the-art methods. Our code will be released.

## 1 Introduction

Continual learning (CL) aims to equip machine learning models with a remarkable capability of human intelligence, *i.e.*, the ability to learn continuously over time, acquiring new knowledge while retaining and integrating prior experience (McCloskey & Cohen, 1989). This capability is crucial for developing general-purpose intelligent systems that can adapt to dynamic and evolving environments without forgetting previously learned information. Traditional CL methods typically operate under the *offline* setting (Parisi et al., 2019), where models have access to the full dataset of each task and are allowed multiple passes over the data. In contrast, *online* continual learning (OCL) presents a more realistic and challenging scenario, where data arrive sequentially in a stream and each sample can be used only once. This streaming constraint closely aligns with practical deployment scenarios such as real-time learning in robotics, edge computing, and user-facing systems (Mai et al., 2022; Gunasekara et al., 2023). In this paper, we focus on the challenging class incremental learning setting of the OCL mode, where the model incrementally learns to recognize all classes observed so far from a data stream in a single pass, and classify test instances without access to task identity.

Experience replay has emerged as the cornerstone of nearly all successful OCL methods (Chaudhry et al., 2019; Guo et al., 2022; Soutif-Cormerais et al., 2023), where a small subset of past data is stored in a memory buffer and interleaved with current samples during training. Earlier approaches imply a *uniform gradient aggregation* strategy and treat gradients from previous tasks equally during joint optimization (Rolnick et al., 2019; Buzzega et al., 2020; Caccia et al., 2022; Wang et al., 2023). However, this simple scheme ignores task heterogeneity, allowing a few dominant tasks to overshadow others and resulting in **unbalanced updates**. *Gradient transformation* approaches modify gradients through projections, constraining gradient updates to be approximately orthogonal to the subspace of past representations to preserve their high-level statistics (Zeng et al., 2019; Saha et al., 2021; Deng et al., 2021; Yoo et al., 2024). However, these methods focus on preserving individual task representations and do not explicitly balance their relative influence, which may result in **uneven retention** of past task knowledge. *Gradient alignment* methods instead reweight gradients according to their conflict with the current task, emphasizing conflicting ones while discarding others (Lopez-Paz & Ranzato, 2017; Chaudhry et al., 2019; Riemer et al., 2019; Guo et al., 2020; Gupta et al., 2020). Though effective at reducing current–past interference, these approaches overlook conflicts among past tasks, which leads to **imbalanced retention**. More recently, Pareto optimization has been explored to capture inter-task relationships more holistically (Wu et al., 2024), but this approach requires complex hyper-gradient computations for stable training.

To tackle the problem of **achieving balanced learning** across past tasks in OCL, we propose FairOCL, a principled framework that mitigates gradient interference in replay-based methods. Our approach builds on an analogy with fair resource allocation in communication networks (Mo & Walrand, 2000). We view each past task as a user competing for resources, where the shared gradient direction serves as the resource to reduce task losses, and the "service quality" of a task is measured by its loss reduction after updating along this direction. Under this perspective, replay-based OCL can be formulated as a utility maximization problem, where each task is assigned an $\alpha$-fair utility function. Different choices of $\alpha$ correspond to different fairness behaviors, among which certain settings coincide with classical notions such as max-min (Radunovic & Le Boudec, 2007), proportional (Kelly, 1997), and minimum potential delay fairness (Kelly et al., 1998). From a CL standpoint, this fairness-driven formulation enhances stability by enforcing fairness among past task gradients to prevent uneven retention, while preserving plasticity by ensuring sufficient gradient contribution from the current task. In addition, to counteract the task recency bias (Chrysakis & Moens, 2023) inherent in replay-based methods, where predictions are biased toward recently observed data, we incorporate knowledge distillation (Hinton et al., 2015) to stabilize feature representations and mitigate drift in earlier tasks.

Our contributions are three-fold: **i)** We provide a fresh perspective that connects replay-based OCL with fair resource allocation in communication networks, framing gradient aggregation as a fairness-driven problem. **ii)** We propose FairOCL, the first framework to formulate gradient aggregation in replay-based OCL as a fairness-constrained utility maximization problem, enabling dynamic and balanced task contributions during training. **iii)** To further mitigate task recency bias inherent in replay-based methods, we incorporate knowledge distillation to stabilize past representations and improve retention. Extensive experiments on standard benchmarks, including CIFAR-10 Krizhevsky et al. (2009), CIFAR-100 Krizhevsky et al. (2009), TinyImageNet Le & Yang (2015), and ImageNet-100 Hou et al. (2019), demonstrate that FairOCL consistently outperforms competitive state-of-the-art baselines across varying buffer sizes, validating both its effectiveness and generality.

## 2 RELATED WORK

**Online continual learning (OCL) and Replay-based Methods.** OCL aims to train models to adapt to non-stationary data streams under a strict single-pass constraint while mitigating catastrophic forgetting (McCloskey & Cohen, 1989). Unlike offline continual learning, which permits multiple passes over task datasets (Parisi et al., 2019), OCL operates under more realistic and restrictive conditions. A foundational strategy in OCL is experience replay (Rolnick et al., 2019), which stores a subset of past data in a memory buffer and interleaves it with incoming samples during training. Recent extensions include DER++ (Buzzega et al., 2020), which distills past experiences from the logits of samples stored in the memory buffer, ER-ACE (Caccia et al., 2022), which introduces asymmetric losses to balance new and replay data, and CBA (Wang et al., 2023), which utilizes stored past samples to correct the recency bias through a bias attractor. Despite their effectiveness, these methods typically aggregate gradients from past and current tasks with uniform weighting, which often produces conflicting gradient directions and ultimately degrades performance.

**Gradient Transformation Methods.** One way to mitigate gradient conflicts is to apply projection-based transformations, where gradient updates are constrained to be orthogonal to the subspace spanned by past representations so as to preserve their high-level statistics. Representative examples include OGD (Zeng et al., 2019) and GPM (Saha et al., 2021), which build orthogonal subspaces to protect important representations of earlier tasks. Later methods improved estimation of projection subspace either by maintaining averaged mini-batch activations to avoid capacity exhaustion (Zeng et al., 2019; Guo et al., 2022) or by employing low-rank approximations of past activations in an SVD-like manner (Saha et al., 2021; Deng et al., 2021; Lin et al., 2022; Saha & Roy, 2023). Recently, LPR (Yoo et al., 2024) extended this line of work to the online setting, introducing a layerwise proximal point formulation that stabilizes replay-based training by encouraging gradual representation updates. Although effective in preserving knowledge, these methods regulate tasks independently without explicitly balancing their relative influences, potentially leading to uneven retention.

**Gradient Alignment Methods.** A complementary line of work mitigates gradient conflicts by adjusting update directions according to their compatibility with past task gradients. GEM (Lopez-Paz & Ranzato, 2017) enforces non-negative inner products between the current gradient update direction and past task gradients, ensuring that new learning does not harm prior knowledge. More efficient variants such as A-GEM (Chaudhry et al., 2019) and MEGA (Guo et al., 2020) reduce

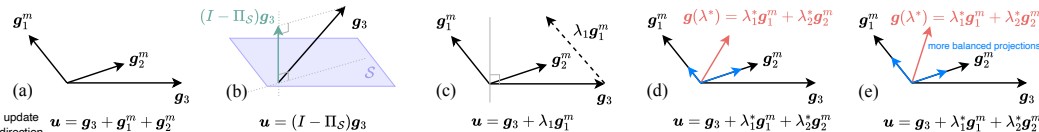

Figure 1: **Conceptual illustration of gradient update strategies.** Here, $\mathcal{S}$ is the subspace spanned by past-task representations. **(a) Uniform gradient aggregation** treat current and past-task gradients equally. **(b) Gradient Transformation** projects $\boldsymbol{g}_3$ onto the subspace orthogonal complement of $\mathcal{S}$ to preserve prior knowledge. **(c) Gradient Alignment** considers $\boldsymbol{g}_1^m$ that conflicts with $\boldsymbol{g}_3$ while neglecting $\boldsymbol{g}_2$ that does not. **(d) POCL** (Wu et al., 2024) obtains weights $\lambda^*$ via Pareto optimization to flexibly manage inter-task trade-offs. **(e) FairOCL (Ours)** formulates gradient aggregation as an $\alpha$-fair utility maximization problem and yields $\lambda^*$ that achieves *more balanced* projections (blue arrows) across tasks. See §3 and §4 for details.

computational cost by averaging historical gradients rather than applying the aforementioned inner project constraint individually. Meta-learning approaches such as MER (Riemer et al., 2019) and La-MAML (Gupta et al., 2020) further adapt update directions through bi-level optimization. Though highlighting the importance of handling gradient conflicts, these methods primarily focus on resolving conflicts between the current task and past tasks, while overlooking relationships among past tasks themselves. More recently, POCL (Wu et al., 2024) employs Pareto optimization to capture inter-task relationships and achieve more holistic gradient balancing, but this approach relies on computationally intensive hyper-gradient calculations for stability. Our method extends this research direction by incorporating fairness-driven constraints, thereby promoting balanced gradients across past tasks without relying on hyper-gradients.

**Fairness in Capacity-Constrained Resource Allocation.** Fair resource allocation has long been studied in wireless communication, where limited resources such as power and bandwidth must be distributed among users in a manner that balances equity and efficiency, and has more recently also appeared in settings like federated learning (Li et al., 2020; Zhang et al., 2022) and multi-task learning (Ban & Ji, 2024). To formalize this trade-off, researchers have proposed a variety of fairness metrics. Jain's fairness index (Jain et al., 1984) quantifies the uniformity of allocations, while proportional fairness (Kelly, 1997) strikes a compromise between equity and throughput by allocating resources in proportion to demand. Max-min fairness (Radunovic & Le Boudec, 2007) instead protects the least-served user by maximizing their minimum allocation. These notions are unified under the $\alpha$-fairness framework (Mo & Walrand, 2000; Lan et al., 2010), where varying $\alpha$ interpolates between utilitarian efficiency ($\alpha! \rightarrow !0$) and max-min equity ($\alpha \rightarrow \infty$). In this paper, we bridge OCL with fair resource allocation theory and propose FairOCL, which formulates gradient aggregation as an $\alpha$-fair utility maximization problem to alleviate gradient interference and mitigate forgetting, while preserving plasticity to learn new tasks.

## 3 PRELIMINARIES

### 3.1 GRADIENT AGGREGATION IN REPLAY-BASED OCL

Online continual learning (OCL) aims to incrementally learn a sequence of $N$ tasks from a non-stationary data stream under strict single-pass constraints. Let $\boldsymbol{g}_n$ denote the gradient of the current task $n$ computed from its incoming batch, where $n = 1, \ldots, N$. Let $\boldsymbol{g}_i^m$ denote the gradient of a previous task $i < n$ computed from stored samples in the memory buffer $\mathcal{M}$.

Traditional replay-based OCL methods (Rolnick et al., 2019; Buzzega et al., 2020; Caccia et al., 2022) calculate the gradient update direction $\boldsymbol{u}$ by aggregating the current gradient $\boldsymbol{g}_n$ with previous task gradients $\boldsymbol{g}_i^m$ through uniform weighting (*cf.*, Fig. 1 (a)):

$$\boldsymbol{u} = \boldsymbol{g}_n + \sum_{i=1}^{n-1} \boldsymbol{g}_i^m. \tag{1}$$

While this *uniform gradient weighting* reduces catastrophic forgetting, it often suffers from gradient conflicts due to its naive averaging scheme. To mitigate such conflicts, *gradient transformation* methods (Zeng et al., 2019; Saha et al., 2021; Deng et al., 2021; Yoo et al., 2024) project the current gradient $\boldsymbol{g}_n$ onto the orthogonal complement of $\mathcal{S}$ to preserve prior representations (*cf.*, Fig. 1 (b)),

where $\mathcal{S}$ is the subspace spanned by past-task representations. *Gradient alignment* methods (Lopez-Paz & Ranzato, 2017; Chaudhry et al., 2019; Guo et al., 2020) instead determine $\boldsymbol{u}$ by solving:

$$\max_{\boldsymbol{u}} \; -\frac{1}{2}\|\boldsymbol{g}_n - \boldsymbol{u}\|_2^2, \quad \text{s.t. } \langle \boldsymbol{u}, \boldsymbol{g}_i^m \rangle \geq 0, \quad i = 1, \ldots, n-1. \tag{2}$$

This formulation ensures that $\boldsymbol{u}$ remains close to the current task gradient $\boldsymbol{g}_n$ to preserve the performance of the current task, while maintaining non-negative alignment with past task gradient $\boldsymbol{g}_i^m$ to prevent negative backward transfer (*cf.*, Fig. 1 (c)). However, the solution implicitly prioritizes past tasks that conflict with the current gradient, while neglecting inter-task relationships.

Building on these formulations, Wu et al. (2024) (*cf.*, Fig. 1 (d)) derived a ***hierarchical gradient aggregation framework***, reformulating the update direction $\boldsymbol{u}$ as:

$$\boldsymbol{u} = \boldsymbol{g}_n + \boldsymbol{g}(\boldsymbol{\lambda}) = \boldsymbol{g}_n + \sum\nolimits_{i=1}^{n-1} \lambda_i \boldsymbol{g}_i^m, \tag{3}$$

where $\boldsymbol{g}_n$ is the current task gradient, and $\boldsymbol{g}(\boldsymbol{\lambda})$ aggregates past gradients with weights $\lambda_i$. This reveals that replay-based methods inherently prioritize the current task gradient, while the aggregation of past task gradients primarily serve as a regularization to mitigate forgetting. Thus, effective OCL requires not only preserving the current task gradient but also fairly balancing past task gradients.

## 3.2 FAIRNESS IN RESOURCE ALLOCATION

An analogous challenge arises in communication networks, where fairness is a central principle for allocating limited resources (*e.g.*, channel bandwidth, transmission rate) among $K$ users. A common approach formulates this as a utility maximization problem under link capacity constraints:

$$\max_{x_1, \ldots, x_K \in C} \sum\nolimits_{i \in [K]} u(x_i) := (1-\alpha)^{-1} x_i^{1-\alpha}, \quad \alpha \in [0,1) \cup (1, +\infty), \tag{4}$$

where $x_i$ denotes the allocation for user $i$ (*e.g.*, transmission rate), $u(\cdot)$ is the utility (*e.g.*, throughput), and $C$ represents the convex set of feasible allocations. The function $(1-\alpha)^{-1} x_i^{1-\alpha}$ is known as the ***α-fair function***. Different choices of $\alpha$ recover widely studied fairness notions. **Proportional fairness** arises as $\alpha \to 1$, where the utility reduces to $\log x_i$, and the objective becomes the sum of logarithmic utilities. **Minimum potential delay fairness** corresponds to $\alpha = 2$, where the objective to maximize $\sum_i -1/x_i$ can be interpreted as minimizing the total delay for transferring unit-size files. **Max-min fairness** is obtained as $\alpha \to +\infty$, which protects the least-served user by maximizing their minimum allocation. We refer readers to (Mo & Walrand, 2000) and (Ban & Ji, 2024) for detailed derivations and more discussion. This well-established framework motivates our treatment of gradient aggregation in OCL as a fairness-driven resource allocation problem.

# 4 FAIROCL: FAIR GRADIENT AGGREGATION IN OCL

Inspired by the principles of fair resource allocation, we propose FairOCL, a framework for fairness-aware and adaptive gradient aggregation in OCL. FairOCL addresses the hierarchical gradient aggregation by regulating past-task gradients to ensure more balanced contributions (*cf.*, Fig. 1 (e)).

## 4.1 PROBLEM FORMULATION

Recall the hierarchical formulation in Eq. (3), where each $\boldsymbol{g}_i^m$ denotes the gradient of a past task $i$. Our goal is to construct an update direction $\boldsymbol{d} = \boldsymbol{g}(\boldsymbol{\lambda})$ that fairly aggregates past-task gradients. Following the analogy to communication networks, we interpret $\boldsymbol{g}_i^{m\top} \boldsymbol{d}$ as the utility (loss reduction rate) $x_i$ for task $i$, and larger $\boldsymbol{g}_i^{m\top} \boldsymbol{d}$ indicate more favorable optimization for task $i$.

To promote fairness among $n-1$ past tasks, we adopt the $\alpha$-fair utility function in Eq. (4), and view each past task as a user in network resource allocation. The utility of user $i$ is calculated as $(1-\alpha)^{-1} x_i^{1-\alpha} = (1-\alpha)^{-1} (\boldsymbol{g}_i^{m\top} \boldsymbol{d})^{1-\alpha}$, where $\alpha \in [0,1) \cup (1, +\infty)$. We then formulate:

$$\max_{\boldsymbol{d} \in B_\epsilon} \sum\nolimits_{i=1}^{n-1} \frac{(\boldsymbol{g}_i^{m\top} \boldsymbol{d})^{1-\alpha}}{1-\alpha}, \quad \text{s.t. } \boldsymbol{g}_i^{m\top} \boldsymbol{d} \geq 0, \tag{5}$$

where $B_\epsilon$ is a ball of radius $\epsilon$ centered at the origin, enforcing a bound on the magnitude of the update direction $\boldsymbol{d}$, similar to the link capacity constraints in network resource allocation. The convexity of the constraint set of $\boldsymbol{d}$ ensures the tractability of the problem. This formulation offers a principled and effective perspective on fairness-aware gradient aggregation for OCL, enabling FairOCL to adaptively modulate task contributions while maintaining optimization efficiency under constraints.

---

**Algorithm 1** FairOCL Training Procedure

---

**Require:** Initial student parameters $\boldsymbol{\theta}_0$; initialize teacher parameters as $\boldsymbol{\theta}_0^{\text{tea}} \leftarrow \boldsymbol{\theta}_0$, fairness parameter $\alpha$, memory buffer $\mathcal{M}$, learning rate $\eta$, number of tasks $N$

1: **for** task $n = 1$ to $N$ **do**              ▷ Sequential task learning
2:     Initialize task model: $\boldsymbol{\theta}_n^{(0)} \leftarrow \boldsymbol{\theta}_{n-1}, \boldsymbol{\theta}_n^{\text{tea}(0)} \leftarrow \boldsymbol{\theta}_{n-1}^{\text{tea}}$
3:     **for** each batch $\mathcal{B}_n^{(b)}$ in task $n$ **do**          ▷ Online updates
4:        Compute current task gradient: $\boldsymbol{g}_n \leftarrow \nabla_{\boldsymbol{\theta}} \mathcal{L}(\boldsymbol{\theta}_n^{(b-1)}, \boldsymbol{\theta}_n^{\text{tea}(b-1)}; \mathcal{B}_n^{(b)})$     ▷ Eq. (9)
5:        Compute past task gradients: $[\boldsymbol{g}_1^m, \ldots, \boldsymbol{g}_{n-1}^m] \leftarrow \nabla_{\boldsymbol{\theta}} \mathcal{L}(\boldsymbol{\theta}_n^{(b-1)}, \boldsymbol{\theta}_n^{\text{tea}(b-1)}; \mathcal{M})$    ▷ Eq. (9)
6:        Solve Eq. (7) to obtain weights $\boldsymbol{\lambda} \leftarrow w^*$
7:        Update direction: $\boldsymbol{u} = \boldsymbol{g}_n + \sum_{i=1}^{n-1} \lambda_i \boldsymbol{g}_i^m$
8:        Update student: $\boldsymbol{\theta}_n^{(b)} = \boldsymbol{\theta}_n^{(b-1)} - \eta \boldsymbol{u}$
9:        Update teacher via EMA                       ▷ Eq. (8)
10:       Update memory buffer $\mathcal{M}$ via reservoir sampling (Vitter, 1985)
11:     **end for**
12:     Final task model: $\boldsymbol{\theta}_n \leftarrow \boldsymbol{\theta}_n^{(b)}, \boldsymbol{\theta}_n^{\text{tea}} \leftarrow \boldsymbol{\theta}_n^{\text{tea}(b)}$
13: **end for**

---

### 4.2 OPTIMIZATION METHOD

Since the objective in Eq. (5) is monotonically non-decreasing in $\boldsymbol{d}$, the optimal solution lies on the boundary of $B_\epsilon$. At the optimum, the KKT conditions (Boyd & Vandenberghe, 2004) imply:

$$\sum_i \boldsymbol{g}_i^m (\boldsymbol{g}_i^{m\top} \boldsymbol{d})^\alpha = c\boldsymbol{d}, \quad c > 0. \tag{6}$$

Following (Ban & Ji, 2024), we set $c = 1$ for simplicity. Assume that the task gradients $\{\boldsymbol{g}_i^m\}$ are linearly independent, we can express $\boldsymbol{d}$ as a non-negative combination $\boldsymbol{d} = \sum_i \omega_i \boldsymbol{g}_i^m$. Substituting yields the relation $(\boldsymbol{g}_i^{m\top} \boldsymbol{d})^\alpha = \omega_i$, which gives the system:

$$\boldsymbol{G}^\top \boldsymbol{G} \omega = \omega^{-1/\alpha}, \quad \alpha \neq 0, \tag{7}$$

where $\boldsymbol{G} = [\boldsymbol{g}_1 \cdots \boldsymbol{g}_n]$ is a gradient matrix and the exponent $-1/\alpha$ is applied elementwise. When $\alpha = 0$, this reduces to uniform weighting where $\omega_i = 1$ for all $i$. Following (Ban & Ji, 2024), we solve Eq. (7) by using a constrained nonlinear least squares formulation. Since FairOCL performs a single backward pass over the current batch combined with memory samples, its computational cost does not increase with the number of tasks. The overall training budget remains comparable to other replay-based baselines (see §A.4).

### 4.3 ALLEVIATING FEATURE DRIFT WITH KNOWLEDGE DISTILLATION

While fair gradient aggregation ensures balanced updates across tasks, replay-based OCL still suffers from task recency bias (Chrysakis & Moens, 2023), which can compromise gradient aggregation as it undermines the stability of representations. To mitigate this, we integrate knowledge distillation with an Exponential Moving Average (EMA) teacher (Tarvainen & Valpola, 2017), where the teacher parameters are updated as:

$$\boldsymbol{\theta}_n^{\text{tea}(b)} = \gamma \boldsymbol{\theta}_n^{(b)} + (1 - \gamma) \boldsymbol{\theta}_n^{\text{tea}(b-1)}, \quad n = 1, \ldots, N. \tag{8}$$

Here, $\boldsymbol{\theta}_n^{(b)}$ denotes the student model parameters at task $n$, batch $b$, and $\gamma \in [0, 1)$ is the momentum coefficient, which is set to $0.1$ empirically in our experiments. The overall training loss integrates standard cross-entropy with a distillation term:

$$\mathcal{L}(\boldsymbol{\theta}, \boldsymbol{\theta}^{\text{tea}}; \mathcal{D}) = \mathcal{L}_{\text{CE}}(\boldsymbol{\theta}; \mathcal{D}) + \beta \mathcal{L}_{\text{KL}}(\boldsymbol{\theta}, \boldsymbol{\theta}^{\text{tea}}, \tau; \mathcal{D}), \tag{9}$$

where $\mathcal{D} = \{\mathcal{B}_n^{(b)}, \mathcal{M}\}$ is the current training data, $\mathcal{L}_{\text{CE}}$ is the cross-entropy loss, $\mathcal{L}_{\text{KL}}$ is the KL divergence with temperature $\tau$, and $\beta$ controls the relative contribution of the distillation term. The overall training pipeline is detailed in Alg. 1. At inference, we average the final weights of teacher and student to obtain a stable model estimate following (Tarvainen & Valpola, 2017).

Table 1: **Average Accuracy (Acc)** (%, ↑) on four benchmark datasets with different memory buffer size $|\mathcal{M}|$. Displayed values are the mean and standard deviation computed over 5 runs. See §5.2.

| Dataset | CIFAR-10 | | | CIFAR-100 | | | TinyImageNet | | | ImageNet-100 | | |
|---|---|---|---|---|---|---|---|---|---|---|---|---|
| $|\mathcal{M}|$ | 200 | 500 | 1000 | 1000 | 2000 | 5000 | 2000 | 5000 | 10000 | 2000 | 5000 | 10000 |
| ER (NeurIPS'19) | $46.33_{\pm2.42}$ | $55.73_{\pm2.04}$ | $62.99_{\pm2.10}$ | $23.00_{\pm3.80}$ | $31.55_{\pm1.27}$ | $38.05_{\pm1.08}$ | $12.29_{\pm0.44}$ | $19.64_{\pm1.14}$ | $26.92_{\pm1.89}$ | $19.06_{\pm0.90}$ | $29.74_{\pm1.34}$ | $36.72_{\pm1.09}$ |
| DER++ (NeurIPS'20) | $47.07_{\pm0.97}$ | $55.53_{\pm1.05}$ | $58.51_{\pm0.68}$ | $22.80_{\pm1.80}$ | $25.89_{\pm1.46}$ | $25.71_{\pm2.40}$ | $16.10_{\pm1.23}$ | $18.49_{\pm1.08}$ | $18.06_{\pm1.30}$ | $15.36_{\pm3.04}$ | $19.19_{\pm1.55}$ | $20.48_{\pm4.67}$ |
| DVC (CVPR'22) | $48.08_{\pm4.27}$ | $58.72_{\pm2.03}$ | $61.11_{\pm2.97}$ | $18.66_{\pm2.54}$ | $22.73_{\pm2.90}$ | $28.47_{\pm3.95}$ | $14.04_{\pm0.75}$ | $18.48_{\pm1.49}$ | $17.95_{\pm2.93}$ | $14.54_{\pm5.15}$ | $21.88_{\pm3.45}$ | $28.50_{\pm2.93}$ |
| GSA (CVPR'23) | $48.90_{\pm3.38}$ | $61.45_{\pm1.95}$ | $67.63_{\pm1.24}$ | $29.68_{\pm1.54}$ | $36.96_{\pm0.79}$ | $45.86_{\pm1.89}$ | $19.08_{\pm0.18}$ | $27.22_{\pm0.95}$ | $32.94_{\pm0.75}$ | $24.29_{\pm0.59}$ | $33.47_{\pm1.18}$ | $40.18_{\pm0.93}$ |
| PCR (CVPR'23) | $52.20_{\pm0.66}$ | $60.61_{\pm2.23}$ | $61.66_{\pm13.86}$ | $30.68_{\pm0.81}$ | $38.63_{\pm1.01}$ | $45.27_{\pm0.78}$ | $19.55_{\pm0.54}$ | $27.02_{\pm0.37}$ | $33.34_{\pm0.22}$ | $19.89_{\pm6.24}$ | $31.35_{\pm3.01}$ | $36.99_{\pm4.70}$ |
| POCL[*] (ICML'24) | $35.69_{\pm3.82}$ | $48.08_{\pm3.72}$ | $56.97_{\pm3.88}$ | $16.48_{\pm1.45}$ | $24.84_{\pm0.89}$ | $32.16_{\pm2.44}$ | $11.59_{\pm0.48}$ | $19.45_{\pm1.72}$ | $24.38_{\pm0.98}$ | – | – | – |
| ER-MKD (ICML'24) | $57.54_{\pm2.55}$ | $68.48_{\pm0.92}$ | $74.33_{\pm0.68}$ | $38.50_{\pm0.50}$ | $45.20_{\pm0.20}$ | $52.10_{\pm0.50}$ | $25.03_{\pm0.57}$ | $33.32_{\pm0.50}$ | $39.69_{\pm0.55}$ | $30.67_{\pm0.46}$ | $39.71_{\pm1.07}$ | $44.92_{\pm0.98}$ |
| **FairOCL** | $\mathbf{58.65_{\pm0.85}}$ | $\mathbf{68.71_{\pm0.89}}$ | $\mathbf{74.83_{\pm0.90}}$ | $\mathbf{39.47_{\pm0.34}}$ | $\mathbf{46.53_{\pm1.03}}$ | $\mathbf{53.39_{\pm0.34}}$ | $\mathbf{26.92_{\pm0.38}}$ | $\mathbf{34.95_{\pm0.46}}$ | $\mathbf{40.52_{\pm0.61}}$ | $\mathbf{31.19_{\pm1.78}}$ | $\mathbf{41.18_{\pm1.55}}$ | $\mathbf{47.22_{\pm0.93}}$ |

[*] denotes results obtained by running the official code.

# 5 EXPERIMENTS

## 5.1 EXPERIMENTAL SETUPS

**Datasets.** Following (Michel et al., 2024; Wu et al., 2024), we consider four representative benchmarks for evaluating OCL. *CIFAR-10* (Krizhevsky et al., 2009) consists of 10 classes with 50,000 training and 10,000 test images, which is partitioned into five disjoint tasks with two classes each. *CIFAR-100* (Krizhevsky et al., 2009) includes 100 classes with the same number of training and test images as CIFAR-10, which is organized into 10 disjoint tasks of 10 classes each. *TinyImageNet* (Le & Yang, 2015) contains 200 classes, with 100,000 training and 10,000 test images, which is split into 20 non-overlapping tasks of 10 classes each. *ImageNet-100* (Hou et al., 2019) is a 100-class subset of ImageNet-1K (Deng et al., 2009), which is divided into 20 tasks, each comprising 5 classes.

**Baselines.** To contextualize the performance of our method, we compare it against a diverse set of state-of-the-art replay-based OCL baselines. ER (Rolnick et al., 2019) is a foundational experience replay method that trains the model using cross-entropy loss on a combination of current and buffered samples. DER++ (Buzzega et al., 2020) enhances ER by introducing a logit-level consistency regularization via knowledge distillation. DVC (Gu et al., 2022) augments replay with a dual-view contrastive loss that maximizes mutual information between different augmentations of input data. GSA (Guo et al., 2023) deal with cross-task class discrimination caused by data imbalance by introducing a self-adaptive loss based on gradients. PCR (Lin et al., 2023) adopts a proxy-based contrastive loss to enhance discriminative feature learning across tasks. POCL (Wu et al., 2024) incorporates Pareto optimization to balance trade-offs among tasks with a hyper-gradient-based implementation. ER-MKD is a basic instantiation of MKD-OCL (Michel et al., 2024), which incorporates a momentum-based knowledge distillation strategy into ER, producing a simple yet competitive baseline. We adopt ER-MKD as it allows us to evaluate the core mechanism of MKD-OCL in isolation from additional complexities introduced when combining it with stronger baselines.

**Evaluation Metrics.** We adopt two standard metrics in the continual learning literature: Average Accuracy and Average Forgetting. Let $\boldsymbol{A}_{i,j}$ denote the test accuracy on task $i$ after training on task $j$. *Average Accuracy (Acc)* measures the overall model performance, and is computed as the average accuracy after learning all the $N$ tasks: $Acc = \frac{1}{N} \sum_{i=1}^{N} \boldsymbol{A}_{i,N}$. *Average Forgetting (F)* quantifies forgetting of previously learned tasks, which is computed as the average of the maximum performance drop of each task after learning the final task: $F = \frac{1}{N-1} \sum_{i=1}^{N-1} \max_{t \in \{1,...,N-1\}} (\boldsymbol{A}_{i,t} - \boldsymbol{A}_{i,N})$. Together, they comprehensively evaluate both plasticity (via Acc) and stability (via F), providing a balanced view of continual learning performance.

**Implementation Details.** Following (Guo et al., 2023; Michel et al., 2024), we use a standard ResNet-18 (He et al., 2016) (not pre-trained) as the backbone for all methods. The memory buffer is maintained using reservoir sampling (Vitter, 1985). Our method is trained using Adam (Kingma & Ba, 2015) with an initial learning rate of 0.0005. The input batch size is set to 10 for CIFAR-10, and 16 for all other datasets. For replay, the memory sampling batch size is $64/80/160$ for CIFAR-10 with $|\mathcal{M}| = 200/500/1000$, 256 for CIFAR-100 and TinyImageNet, and 128 for ImageNet-100. Each experiment is run on a single NVIDIA GeForce RTX 4090 GPU. The fairness parameter $\alpha$ is set to 0.5 for CIFAR-10, and 1.5 for all other three datasets. For knowledge distillation, we use a temperature $\tau = 3.0$ and a distillation loss weight $\beta = 5.5$.

Table 2: **Average Forgetting (F)** (%, ↓) on four benchmark datasets with different memory buffer size $|\mathcal{M}|$. Displayed values are the mean and standard deviation computed over 5 runs. See §5.2.

| Dataset | CIFAR-10 | | | CIFAR-100 | | | TinyImageNet | | | ImageNet-100 | | |
|---|---|---|---|---|---|---|---|---|---|---|---|---|
| $|\mathcal{M}|$ | 200 | 500 | 1000 | 1000 | 2000 | 5000 | 2000 | 5000 | 10000 | 2000 | 5000 | 10000 |
| ER (NeurIPS'19) | $52.90_{\pm3.54}$ | $34.07_{\pm3.13}$ | $23.43_{\pm3.41}$ | $33.04_{\pm1.75}$ | $22.84_{\pm1.27}$ | $14.55_{\pm2.73}$ | $40.68_{\pm0.72}$ | $28.43_{\pm1.14}$ | $20.78_{\pm1.00}$ | $43.69_{\pm2.12}$ | $28.04_{\pm1.11}$ | $0_{\pm3.37}$ |
| DER++ (NeurIPS'20) | $36.93_{\pm5.86}$ | $28.26_{\pm5.06}$ | $19.47_{\pm3.20}$ | $33.01_{\pm4.85}$ | $29.31_{\pm3.65}$ | $30.10_{\pm3.56}$ | $32.28_{\pm7.37}$ | $33.56_{\pm3.80}$ | $33.42_{\pm4.83}$ | $27.68_{\pm1.04}$ | $17.53_{\pm2.51}$ | $19.04_{\pm0.57}$ |
| DVC (CVPR'22) | $45.36_{\pm4.08}$ | $30.64_{\pm2.16}$ | $24.21_{\pm3.18}$ | $41.24_{\pm2.28}$ | $36.20_{\pm2.54}$ | $33.26_{\pm3.13}$ | $38.20_{\pm2.49}$ | $39.01_{\pm5.46}$ | $40.71_{\pm7.99}$ | $53.97_{\pm2.25}$ | $41.88_{\pm4.36}$ | $34.86_{\pm4.66}$ |
| GSA (CVPR'23) | $35.85_{\pm4.05}$ | $23.74_{\pm2.47}$ | $16.93_{\pm2.58}$ | $30.65_{\pm0.61}$ | $20.78_{\pm1.89}$ | $8.15_{\pm0.37}$ | $36.98_{\pm0.55}$ | $22.72_{\pm0.92}$ | $13.66_{\pm0.84}$ | $48.30_{\pm1.26}$ | $32.14_{\pm0.82}$ | $19.15_{\pm0.81}$ |
| PCR (CVPR'23) | $\mathbf{30.90}_{\pm4.50}$ | $26.92_{\pm5.01}$ | $19.57_{\pm3.35}$ | $34.45_{\pm1.67}$ | $25.41_{\pm1.71}$ | $15.03_{\pm1.97}$ | $27.88_{\pm2.10}$ | $16.86_{\pm0.60}$ | $9.78_{\pm0.64}$ | $37.93_{\pm1.84}$ | $27.17_{\pm3.37}$ | $19.04_{\pm2.38}$ |
| POCL* (ICML'24) | $55.01_{\pm4.87}$ | $34.82_{\pm6.14}$ | $24.78_{\pm4.92}$ | $31.73_{\pm0.66}$ | $19.31_{\pm1.06}$ | $11.94_{\pm2.12}$ | $36.04_{\pm0.84}$ | $20.18_{\pm2.20}$ | $13.53_{\pm1.56}$ | – | – | – |
| ER-MKD (ICML'24) | $39.95_{\pm3.43}$ | $21.48_{\pm1.38}$ | $\mathbf{10.96}_{\pm1.73}$ | $16.62_{\pm1.10}$ | $10.73_{\pm0.89}$ | $\mathbf{5.31}_{\pm1.20}$ | $25.21_{\pm1.25}$ | $16.49_{\pm0.97}$ | $10.02_{\pm0.97}$ | $25.22_{\pm2.54}$ | $17.00_{\pm1.94}$ | $11.91_{\pm0.57}$ |
| **FairOCL** | $32.11_{\pm2.00}$ | $\mathbf{20.82}_{\pm1.77}$ | $13.16_{\pm0.64}$ | $\mathbf{15.80}_{\pm1.40}$ | $\mathbf{10.71}_{\pm1.65}$ | $6.51_{\pm0.98}$ | $\mathbf{19.40}_{\pm1.09}$ | $\mathbf{14.00}_{\pm0.79}$ | $\mathbf{9.80}_{\pm0.81}$ | $\mathbf{23.46}_{\pm1.54}$ | $\mathbf{14.83}_{\pm1.16}$ | $\mathbf{11.03}_{\pm0.91}$ |

* denotes results obtained by running the official code.

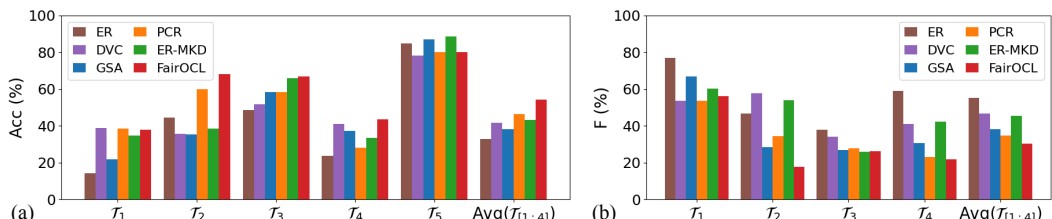

Figure 2: (a) Average Accuracy (Acc) and (b) Average Forgetting (F) of different OCL methods on CIFAR-10 with $N=5$ and buffer size $|\mathcal{M}|$ as 200. Here, $\mathcal{T}_1$-$\mathcal{T}_4$ denotes the previously learned tasks, $\mathcal{T}_5$ is the current task, and $\text{Avg}(\mathcal{T}_{[1:4]})$ means the average performance of past tasks. See §5.3.

## 5.2 COMPARISON RESULTS

**Average Accuracy (Acc) Results.** Table 1 reports the mean and standard deviation of Average Accuracy (Acc) over 5 runs on four benchmarks with varying buffer sizes $|\mathcal{M}|$. FairOCL consistently achieves the best performance across all settings. On CIFAR-10, it yields $58.65\%$, $68.71\%$, and $74.83\%$ at buffer sizes 200, 500, and 1000, corresponding to relative improvements of $1.93\%$, $0.34\%$, and $0.67\%$ over the strongest baseline ER-MKD. On CIFAR-100, FairOCL reaches $39.47\%$, $46.53\%$, and $53.39\%$ for buffer sizes 1000, 2000, and 5000, outperforming ER-MKD by $3.22\%$, $2.94\%$, and $2.48\%$. On TinyImageNet, it achieves $26.92\%$, $34.95\%$, and $40.52\%$ with buffer sizes 2000, 5000, and 10000, surpassing ER-MKD by $7.55\%$, $4.89\%$, and $2.09\%$. Finally, on ImageNet-100, FairOCL obtains $31.19\%$, $41.18\%$, and $47.22\%$ under buffer sizes 2000, 5000, and 10000, with relative gains of $1.70\%$, $3.70\%$, and $5.12\%$. These results demonstrate that FairOCL achieves robust improvements across both simpler and more challenging benchmarks, highlighting its effectiveness in balancing learning and retention. Runtime analysis in §A.4 confirms that FairOCL remains comparable in cost to other replay-based methods.

**Average Forgetting (F) Results.** Table 2 presents Average Forgetting (F) across the same benchmarks, where lower values indicate better knowledge retention. FairOCL consistently achieves the lowest or near-lowest forgetting across most buffer sizes, confirming its resistance to catastrophic forgetting. On CIFAR-10 and CIFAR-100, it performs best in low-resource regimes, while slightly trailing ER-MKD at larger buffer sizes, though with narrow margins. More importantly, on the more challenging TinyImageNet and ImageNet-100, FairOCL delivers the lowest forgetting across all buffer sizes. For example, it reduces forgetting on TinyImageNet to $19.40\%$, $14.00\%$, and $9.80\%$ (buffers 2000, 5000, 10000), and on ImageNet-100 to $23.46\%$, $14.83\%$, and $11.03\%$, outperforming ER-MKD and other strong baselines. Overall, these results confirm that FairOCL not only improves accuracy but also provides superior retention in the OCL setting.

## 5.3 VERIFICATION ON FAIROCL

**Past Task Knowledge Preservation.** We first evaluate how well different methods preserve past knowledge by computing Average Accuracy (Acc) on all previously encountered tasks after learning each new task. As shown in Fig. 2 (a), FairOCL maintains consistently high accuracy across tasks, particularly for earlier ones ($\mathcal{T}_1$-$\mathcal{T}_4$). While it is not the best on the most recent task $\mathcal{T}_5$, FairOCL

Table 3: **Mean Average Accuracy (Acc)** on previous tasks $\mathcal{T}_{[1:n-1]}$ and on all tasks $\mathcal{T}_{[1:n]}$ for CIFAR-10 with $|\mathcal{M}| = 200$ and $N = 5$, where $n = 2, \ldots, N$. See §5.3 for details.

| Methods | $n=2$ | | $n=3$ | | $n=4$ | | $n=5$ | |
|---|---|---|---|---|---|---|---|---|
| | Avg($\mathcal{T}_1$) | Avg($\mathcal{T}_{[1:2]}$) | Avg($\mathcal{T}_{[1:2]}$) | Avg($\mathcal{T}_{[1:3]}$) | Avg($\mathcal{T}_{[1:3]}$) | Avg($\mathcal{T}_{[1:4]}$) | Avg($\mathcal{T}_{[1:4]}$) | Avg($\mathcal{T}_{[1:5]}$) |
| ER (NeurIPS'19) | $55.98_{\pm1.97}$ | $72.34_{\pm0.71}$ | $49.36_{\pm6.67}$ | $60.38_{\pm4.81}$ | $39.45_{\pm4.76}$ | $50.62_{\pm3.22}$ | $32.98_{\pm1.80}$ | $43.73_{\pm1.14}$ |
| DVC (CVPR'22) | $49.29_{\pm17.36}$ | $68.28_{\pm6.29}$ | $51.50_{\pm3.78}$ | $60.55_{\pm1.62}$ | $45.12_{\pm2.96}$ | $53.99_{\pm1.19}$ | $39.17_{\pm5.77}$ | $46.97_{\pm3.30}$ |
| GSA (CVPR'23) | $73.05_{\pm10.29}$ | $74.12_{\pm4.11}$ | $58.85_{\pm3.37}$ | $63.36_{\pm2.34}$ | $49.88_{\pm3.28}$ | $57.07_{\pm1.57}$ | $42.53_{\pm4.13}$ | $49.98_{\pm3.20}$ |
| PCR (CVPR'23) | $66.38_{\pm11.02}$ | $75.70_{\pm3.87}$ | $62.31_{\pm7.83}$ | $65.83_{\pm3.01}$ | $54.33_{\pm7.13}$ | $57.42_{\pm4.25}$ | $47.16_{\pm3.88}$ | $52.86_{\pm1.73}$ |
| ER-MKD (ICML'24) | $71.54_{\pm9.25}$ | $80.12_{\pm3.51}$ | $\mathbf{63.09_{\pm3.09}}$ | $\mathbf{70.23_{\pm2.65}}$ | $57.21_{\pm3.39}$ | $63.40_{\pm3.60}$ | $47.34_{\pm2.97}$ | $55.35_{\pm1.91}$ |
| **FairOCL** | $\mathbf{77.43_{\pm7.17}}$ | $\mathbf{81.44_{\pm3.40}}$ | $62.82_{\pm4.09}$ | $69.54_{\pm2.74}$ | $\mathbf{59.14_{\pm3.97}}$ | $\mathbf{63.95_{\pm2.63}}$ | $\mathbf{52.94_{\pm2.09}}$ | $\mathbf{58.65_{\pm0.85}}$ |

Figure 3: Task-based confusion matrix of various gradient-alignment-based OCL methods on the CIFAR-10 dataset with $|\mathcal{M}| = 200$. See §5.3 for details.

achieves the highest average accuracy over past tasks (Avg($\mathcal{T}_{[1:4]}$)), demonstrating its ability to retain historical knowledge and avoid overfitting to the most recent task. In addition, Fig. 2 (b) shows the Average Forgetting (F) per task. FairOCL consistently exhibits lower forgetting across all tasks, with strong performance on Avg($\mathcal{T}_{[1:4]}$), underscoring its ability to mitigate catastrophic forgetting and ensure preservation of earlier task knowledge.

Table 3 further details accuracy on past tasks (Avg($\mathcal{T}_{[1:n-1]}$)) and all tasks (Avg($\mathcal{T}_{[1:n]}$)) as the task sequence grows from $n = 2$ to $n = 5$. FairOCL consistently surpasses all baselines in past-task knowledge retention. For instance, at $n = 5$, it achieves 52.94% on past tasks and 58.65% overall, compared to ER-MKD (47.34% / 55.35%) and PCR (46.13% / 52.86%). This demonstrates that FairOCL's fairness-driven aggregation directly translates into improved overall performance.

**Task Interrelationships Analysis.** Beyond average accuracy on past tasks, we also examine task interrelationships using the confusion matrices. An effective method should display strong diagonal values, indicating accurate performance on each task, and low off-diagonal entries, reflecting limited interference across tasks. As shown in Fig. 3, FairOCL achieves the most uniform and elevated diagonal values, particularly on earlier tasks $\mathcal{T}_1 - \mathcal{T}_4$, showing that it preserves knowledge more evenly throughout the task sequence. In contrast, ER, ER-MKD, and GSA all exhibit stronger bias toward the most recent task $\mathcal{T}_5$ while showing weaker retention of earlier tasks. Overall, the inter-task analysis confirms that FairOCL maintains more balanced knowledge preservation across tasks, mitigating forgetting while sustaining strong performance on current tasks.

## 5.4 Ablation Study

**Component Analysis.** To evaluate the contribution of each component in FairOCL, we perform an ablation study on CIFAR-10 ($|\mathcal{M}| = 1000$), CIFAR-100 ($|\mathcal{M}| = 2000$), and TinyImageNet ($|\mathcal{M}| = 5000$). As shown in Table 4, both the fair gradient aggregation strategy ("Fair") and the recency-bias mitigation via knowledge distillation ("RB-KD") contribute substantially, but serve complementary purposes. Fairness alone delivers consistent improvements across datasets, confirming its role in balancing past-task contributions and alleviating gradient interference, an aspect that RB-KD cannot address. While RB-KD yields larger standalone improvements in some cases, it mainly stabilizes representations against drift and does not address imbalanced gradient updates. The best performance is consistently obtained when both components are combined, indicating that fairness provides the principled foundation for balanced continual learning, while RB-KD acts as an auxiliary stabilizer that further enhances retention. Together, they form a synergistic framework that surpasses either component in isolation and is essential to the full effectiveness of FairOCL.

Table 4: **Component analysis on FairOCL.** Here, "Fair" denotes the fair gradient aggregation strategy, and "RB-KD" denotes the task recency bias alleviation via knowledge distillation. Results show that both components contribute complementary benefits, and the best performance is consistently achieved when both are combined. See §5.4 for details.

| Fair | RB-KD | CIFAR-10 ($|\mathcal{M}|=1000$) | | CIFAR-100 ($|\mathcal{M}|=2000$) | | TinyImageNet ($|\mathcal{M}|=5000$) | |
|---|---|---|---|---|---|---|---|
| | | Acc ($\uparrow$) | F ($\downarrow$) | Acc ($\uparrow$) | F($\downarrow$) | Acc ($\uparrow$) | F($\downarrow$) |
| ✗ | ✗ | 56.84 | 38.88 | 31.21 | 23.14 | 19.64 | 29.55 |
| ✓ | ✗ | 62.00 | 27.66 | 40.00 | 17.08 | 26.88 | 20.82 |
| ✗ | ✓ | 67.95 | 20.73 | 44.78 | 11.21 | 33.16 | 16.05 |
| ✓ | ✓ | **68.44** | **19.38** | **46.43** | **9.02** | **34.51** | **14.08** |

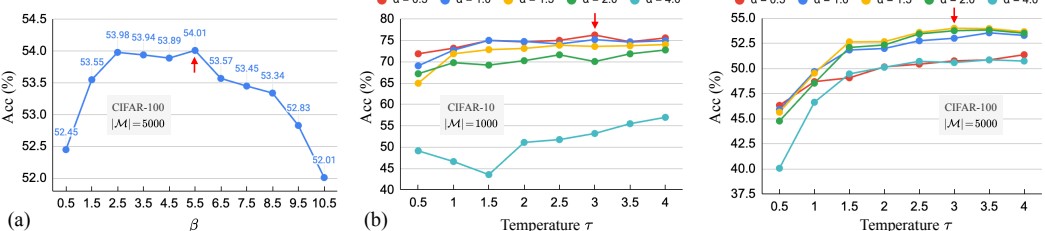

Figure 4: **Ablation study on hyperparameters. (a) Effect of loss weight** $\beta$ on CIFAR-100 ($|\mathcal{M}| = 5000$). **(b) Effect of fairness parameter** $\alpha$ **and temperature** $\tau$ on CIFAR-10 ($|\mathcal{M}| = 1000$) and CIFAR-100 ($|\mathcal{M}| = 5000$). FairOCL shows stable performance across a broad range of $\alpha$ values, indicating robustness to this parameter. See §5.4 for details.

**Hyperparameters.** We first determine the loss weight $\beta$ in Eq. (9). Fig. 4 (a) reports the Average Accuracy (Acc) on CIFAR-100 with $|\mathcal{M}|=5000$ under different values of $\beta$. The best performance is obtained at $\beta=5.5$, which we adopt as the default for all experiments.

We then study the effects of the fairness parameter $\alpha$ and the distillation temperature $\tau$. Fig. 4 (b) shows results on CIFAR-10 with $|\mathcal{M}| = 1000$ (left) and CIFAR-100 with $|\mathcal{M}| = 5000$ (right). On CIFAR-10, the best performance occurs at $\alpha = 0.5$ and $\tau = 3.0$, which we use for all CIFAR-10 settings. On CIFAR-100, the optimal configuration is $\alpha = 1.5$, $\tau = 3.0$, which also performs well on ImageNet-100 and TinyImageNet, and we apply it directly to these two datasets without further tuning. More broadly, Fig. 4 (b) shows that FairOCL is robust across a wide range of $\alpha$ values, with performance varying only gradually. Thus, $\alpha$ can be treated as a principled control parameter for fairness preferences rather than a fragile hyperparameter requiring extensive search.

# 6 CONCLUSION

This study targets a key limitation of current replay-based online continual learning (OCL) methods, namely, the gradient conflicts that arise when jointly training on mixed-task data, which hinder the retention of previously acquired knowledge. Our proposed FairOCL framework mitigates this challenge by formulating gradient aggregation as a constrained utility maximization problem inspired by fair resource allocation principles in communication networks. This formulation promotes fair and adaptive task-level gradient updates, thereby effectively alleviating interference among tasks. To further mitigate the task-recency bias caused by class imbalance, we incorporate a knowledge distillation mechanism that stabilizes representations of earlier tasks. Extensive experiments on standard OCL benchmarks show that FairOCL consistently outperforms state-of-the-art baselines, establishing it as a principled and effective solution for OCL.

**Limitations and Future Work.** While FairOCL shows competitive results in balancing stability and plasticity through gradient fairness, future work could explore mechanisms to automatically adapt the fairness parameter $\alpha$ during training. Another promising direction is to extend the fairness-based formulation to offline continual learning, where multiple passes over data are possible, to further assess its benefits under less restrictive conditions.

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

# A APPENDIX

In this section, we provide additional information to support the main content of the paper. §A.1 documents our use of large language models (LLMs). §A.2 presents a toy multi-objective example to illustrate the effect of different fairness parameters. §A.3 offers a detailed derivation of Eq. (6). §A.4 reports the computational efficiency of FairOCL compared to other baseline methods. Finally, §A.5 analyzes the effect of class coverage in replay batches.

## A.1 USE OF LARGE LANGUAGE MODELS (LLMS)

We used LLMs only for polishing the text, specifically, improving grammar, wording, clarity, and fluency. We did not use LLMs for generating core scientific content, defining the methodology, deriving proofs, designing experiments, interpreting results, or any other technical components. We take full responsibility for correctness, factual claims, and attribution.

## A.2 ILLUSTRATION OF DIFFERENT FAIRNESS PARAMETERS

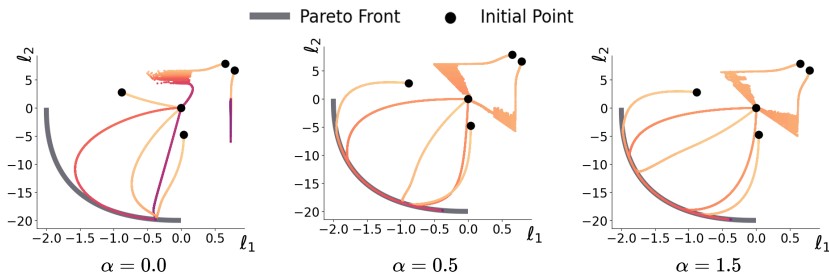

Figure 5: Illustrative of convergence for a two-objective optimization example. Here shows trajectories (color gradient from orange to purple) from five distinct initializations (black dots ● ) under varying fairness parameters $\alpha \in \{0.0, 0.5, 1.5\}$. The Pareto front is highlighted, demonstrating how $\alpha$ affects the trade-off between objectives $L_1$ and $L_2$. See §A.2 for details.

To demonstrate the effect of the fairness parameter $\alpha$, we adopt a two-objective optimization problem from (Ban & Ji, 2024), with two optimization objectives $L_1(\boldsymbol{x})$ and $L_2(\boldsymbol{x})$ are defined for $\boldsymbol{x} = (x_1, x_2)^\top \in \mathbb{R}^2$:

$$L_1(\boldsymbol{x}) = 0.1 \cdot (f_1(\boldsymbol{x})g_1(\boldsymbol{x}) + f_2(\boldsymbol{x})h_1(\boldsymbol{x})), \tag{10}$$
$$L_2(\boldsymbol{x}) = f_1(\boldsymbol{x})g_2(\boldsymbol{x}) + f_2(\boldsymbol{x})h_2(\boldsymbol{x}), \tag{11}$$

where the component functions are given by:

$$f_1(\boldsymbol{x}) = \max(\tanh(0.5x_2), 0), \tag{12}$$
$$f_2(\boldsymbol{x}) = \max(\tanh(-0.5x_2), 0), \tag{13}$$
$$g_1(\boldsymbol{x}) = \log\left(\max(|0.5(-x_1 - 7) - \tanh(-x_2)|, 0.000005)\right) + 6, \tag{14}$$
$$g_2(\boldsymbol{x}) = \log\left(\max(|0.5(-x_1 + 3) - \tanh(-x_2) + 2|, 0.000005)\right) + 6, \tag{15}$$
$$h_1(\boldsymbol{x}) = ((-x_1 + 7)^2 + 0.1(-x_1 - 8)^2)/10 - 20, \tag{16}$$
$$h_2(\boldsymbol{x}) = ((-x_1 - 7)^2 + 0.1(-x_1 - 8)^2)/10 - 20. \tag{17}$$

The gradient magnitude disparity ($\| L_2(\boldsymbol{x}) \| > \| L_1(\boldsymbol{x}) \|$) poses challenges for multi-objective balancing. We investigate five initial points $\{(-8.5, 7.5), (0, 0), (9.0, 9.0), (-7.5, -0.5), (9.0, -1.0)\}$ using Adam optimizer with a learning rate of $1 \times 10^{-3}$. The training process stops when the Pareto front is reached. Fig. 5 shows the optimization trajectories under different fairness regimes. When $\alpha = 0$ (*i.e.*, without fairness), the algorithm may fail to converge to a Pareto stationary point. In contrast, settings with fairness ($\alpha > 0$) lead to convergence to the Pareto front, with the degree of fairness shaped by the choice of $\alpha$. Specifically, $\alpha = 0.5$ is used in our CIFAF-10 experiments, while $\alpha = 1.5$ is adopted for CIFAR-100, TinyImageNet and ImageNet-100. These results demonstrate that incorporating fairness not only ensures convergence but also allows for flexible control over fairness behavior across tasks.

## A.3 Derivation of Eq. (6)

We first show that the optimal solution lies on the boundary of the Euclidean ball $B_\epsilon$, *i.e.*, $||\boldsymbol{d}||_2 = \epsilon$. This follows from the fact that the objective is monotonically increasing in the norm of $\boldsymbol{d}$, under the constraint $\boldsymbol{g}_i^\top \boldsymbol{d} \geq 0$ for all $i$. Specifically:

- The objective in Eq. (5) is a sum of terms of the form $\frac{(\boldsymbol{g}_i^\top \boldsymbol{d})^{1-\alpha}}{1-\alpha}$.

- If $\alpha < 1$, the function $(\boldsymbol{g}_i^\top \boldsymbol{d})^{1-\alpha}$ is a monotonically increasing in $\boldsymbol{g}_i^\top \boldsymbol{d}$ (since the exponent $1-\alpha > 0$).

- If $\alpha > 1$, the term $\frac{(\boldsymbol{g}_i^\top \boldsymbol{d})^{1-\alpha}}{1-\alpha}$ can be rewritten as $-\frac{(\boldsymbol{g}_i^\top \boldsymbol{d})^{1-\alpha}}{\alpha-1}$, which is also a monotonically increasing function of $\boldsymbol{g}_i^\top \boldsymbol{d}$ (since the exponent $1-\alpha < 0$, but the negative sign flips the monotonicity).

Since $\boldsymbol{g}_i^\top \boldsymbol{d}$ is non-decreasing with $||\boldsymbol{d}||_2$ (due to the constraint), the objective is non-decreasing in $\boldsymbol{d}$, and is therefore maximized when $||\boldsymbol{d}||_2 = \epsilon$.

Next, to derive Eq. (6), we apply the Karush-Kuhn-Tucker (KKT) conditions (Boyd & Vandenberghe, 2004) to the constrained optimization in Eq. (5). The Lagrangian is:

$$\mathcal{L}(\boldsymbol{d}, c, \{\mu_i\}) = \sum_{i=1}^{n-1} \frac{(\boldsymbol{g}_i^\top \boldsymbol{d})^{1-\alpha}}{1-\alpha} - \frac{c}{2}(\boldsymbol{d}^\top \boldsymbol{d} - \epsilon^2) + \sum_{i=1}^{n-1} \mu_i(\boldsymbol{g}_i^\top \boldsymbol{d}). \tag{18}$$

The KKT conditions yield:

- **Stationarity**: $\nabla_d \mathcal{L} = \sum_{i=1}^{n-1}(\boldsymbol{g}_i^\top \boldsymbol{d})^{-\alpha}\boldsymbol{g}i - c\boldsymbol{d} + \sum_{i=1}^{n-1} \mu_i\boldsymbol{g}_i = 0$.
  Rearranging gives: $\sum_{i=1}^{n-1}[(\boldsymbol{g}_i^\top \boldsymbol{d})^{-\alpha} + \mu_i]\boldsymbol{g}_i = c\boldsymbol{d}$.
- **Primal Feasibility**: $||\boldsymbol{d}||_2^2 \leq \epsilon^2$ and $\boldsymbol{g}_i^\top \boldsymbol{d} \geq 0$ .
- **Dual Feasibility**: $c \geq 0, \mu_i \geq 0$.
- **Complementary Slackness**: $c(||\boldsymbol{d}||_2^2 - \epsilon^2) = 0, \mu_i(\boldsymbol{g}_i^\top \boldsymbol{d}) = 0, \forall i$.

At the optimum, $||\boldsymbol{d}||_2 = \epsilon$, implying $c > 0$.

Since $\boldsymbol{g}_i^\top \boldsymbol{d} \geq 0$, we get $\mu_i = 0$ for all $i$.

Substituting into the stationarity condition gives: $\sum_{i=1}^{n-1}((\boldsymbol{g}_i^\top \boldsymbol{d})^{-\alpha})\boldsymbol{g}_i = c\boldsymbol{d}, c > 0$, which is Eq. (6).

## A.4 Computational Efficiency

To assess the computational efficiency of FairOCL, we compare its training time with five replay-based OCL methods, namely, DVC (Gu et al., 2022), GSA (Guo et al., 2023), PCR (Lin et al., 2023), POCL (Wu et al., 2024), and ER-MKD (Michel et al., 2024). All experiments are conducted on CIFAR-10 with a buffer size of $\mathcal{M} = 200$. As shown in Table 5, FairOCL achieves the highest average accuracy (58.65%) while maintaining competitive training time (545.2s). Notably, FairOCL is $3\times$ faster than the POCL (1628.2s), which relies on computationally intensive hyper-gradient optimization. Moreover, FairOCL exhibits similar or better efficiency compared to DVC (552.2s) and PCR (547.4s), despite outperforming them in terms of accuracy. These results show that FairOCL delivers strong performance without incurring additional computational overhead.

Table 5: Comparison of training time and average accuracy for different OCL methods on CIFAR-10 ($\mathcal{M} = 200$). See §A.4 for details.

| Method | DVC | GSA | PCR | POCL | ER-MKD | FairOCL |
|---|---|---|---|---|---|---|
| Average Accuracy ($\%, \uparrow$) | 48.08 | 48.90 | 52.20 | 35.69 | 57.54 | **58.65** |
| Training Time (s) | 552.2 | 400.6 | 547.4 | 1628.2 | 355.4 | 545.2 |

## A.5 EFFECT OF CLASS COVERAGE

FairOCL does not explicitly enforce the presence of all previously seen classes in each replay batch. To evaluate whether such coverage improves performance, we conducted experiments on CIFAR-10. As shown in Table 6, enforcing class coverage did not lead to noticeable accuracy gains, but in our cases, slightly degraded performance. This may be because enforcing such class coverage can disrupt the natural sampling distribution maintained by reservoir sampling. More importantly, our method focuses on mitigating *task-level* interference. Although individual classes may be temporarily absent from a batch, they are unlikely to remain absent across many consecutive batches, and it is also rare for all classes from a task to be absent simultaneously. Thus, FairOCL can still compute meaningful task-level gradients and apply fairness-aware updates effectively. Over time, most classes can be sampled with sufficient frequency to support stable learning dynamics. Experimental results in Fig. 2 confirm that FairOCL can mitigate task interference effectively without requiring strict class coverage in each batch compared with other methods.

Table 6: Effect of class coverage. Here presents the Average Accuracy (Acc, %, ↑) on CIFAR-10 with different buffer sizes $|\mathcal{M}|$. See §A.5 for details.

| Class Coverage? | $|\mathcal{M}| = 200$ | $|\mathcal{M}| = 500$ | $|\mathcal{M}| = 1000$ |
|---|---|---|---|
| Yes | 57.41 | 69.43 | 74.12 |
| **No** | **59.22** | **69.94** | **76.26** |

