# OpenReview forum: "FairOCL: Fair Gradient Aggregation for Online Continual Learning"
_ICLR.cc/2026/Conference — ICLR 2026 Conference Withdrawn Submission_

### Official Review · Reviewer_fPYP · 2025-10-19

**Soundness:** 3
**Presentation:** 2
**Contribution:** 2
**Rating:** 2
**Confidence:** 5

**Summary:**

The paper consider the online continual learning, where the data only appears once. The authors proposed the fairness-based gradient alignment approach to alleviate the catastrophic forgetting.

**Strengths:**

1. The authors conducted experiments on various dataset, including two challenging datasets, TinyImageNet and ImageNet-100.

**Weaknesses:**

1. Although the proposed algorithm appears technically sound, the paper’s contribution seems incremental, as it primarily combines two existing methods (i.e., FairGrad and MKD) and closely resembles FairGrad in both structure and writing style. For example, Section 4.1 in FairOCL is almost identical to Section 4 in FairGrad, Section 3.2 in FairOCL mirrors Section 3.2 in FairGrad, and the method described in Section 4.2 of FairOCL is highly similar to that in Section 4.2 of FairGrad.
2. The authors claim that prior works rely on computationally intensive hyper-gradient calculations for stability, whereas their proposed method avoids this issue. However, no concrete justification or empirical evidence supporting this claim is provided.
3. The literature review omits several important works in continual learning, particularly those focused on gradient-based approaches, which limits the paper’s ability to position its contribution within the broader research context.
4. The EMA/MKD method itself is a well-known and effective technique for continual learning. It would strengthen the paper if the authors discussed how FairOCL improves upon or differs from the conventional EMA-based approaches.
5. Since the proposed method is based on gradient alignment, it would be valuable for the authors to provide empirical observations or visualizations illustrating how gradient alignment behaves in practical experiments.
6. The results presented in Table 1 appear to be directly copied from Table 3 of [R1]
    - [R1]: Rethinking momentum knowledge distillation in online continual learning
7. The reported results of average forgetting in Table 2 differ substantially from those presented in the original manuscripts of the compared methods. The authors should carefully discuss the reasons behind these discrepancies. For example, the results for POCL, DER++, ER, and GSA vary significantly from Table 3 in the referenced paper. Moreover, the results of POCL on ImageNet-100 are missing without explanation.
8. The experimental comparisons seem unfair. Given that FairOCL combines FairGrad and MKD, it should be compared against similarly enhanced baselines. Many existing methods could also integrate MKD to improve performance, so limiting comparisons to naive baselines weakens the validity of the claimed advantages.
9. Finally, as the paper emphasizes computational efficiency and gradient-based learning, it would be beneficial to include comparisons of both accuracy and computational cost with other gradient-based methods to substantiate the claims made in the introduction.

**Questions:**

1. The plots in Figure 5 for $\alpha = 0.0$ and $\alpha = 1.5$ appear almost identical to Figures 1(a) and 1(b) in the FairGrad paper, despite the differing experimental setups (multi-task learning and continual learning). The authors should clarify this similarity.

---

### Official Review · Reviewer_YgXt · 2025-10-28

**Soundness:** 2
**Presentation:** 2
**Contribution:** 2
**Rating:** 2
**Confidence:** 3

**Summary:**

The paper introduces a principled and tunable approach to fair continual learning by casting gradient aggregation as an α-fair utility maximization problem, drawing inspiration from network resource allocation.

**Strengths:**

This framing provides a theoretical knob to balance plasticity and stability.
It also connects fairness directly to the optimization dynamics.
The inclusion of a teacher-student EMA distillation mechanism to mitigate feature drift is a reasonable practical addition to help stabilize representations without significantly increasing memory requirements.

**Weaknesses:**

1. My main concerns is whether the particular way of formulating this as a utility maximization problem makes more sense than the classical ERM formulation. The Equation 7 seems ill posed. Specifically, G^\top G \omega=\omega^{-1/\alpha} is the stationarity condition of a non-convex problem.  Existence/uniqueness requires symmetric positive definite G^\top G and strictly positive \omega; in practice G^\top G is often singular/ill-conditioned, and \omega^{-1/\alpha} is undefined at \omega_i\le 0. Even when relaxed to least-squares, it’s solving a noisy non-linear system that may have multiple or no solutions and is highly initialization-sensitive.

2. It's not clear whether the new updates will not harm past tasks as the fairness weighting is a soft heuristic. This is unlike gradient projection or gradient alignment techniques that explicitly constrain the updates.

3. Sensitivity to large gradient norms: The fairness mechanism seems to be dependent on gradient magnitudes. It's possible tasks with large gradient norms can dominate the ones with smaller norms. So is the fairness scale dependent?

**Questions:**

It's not clear whether the new updates will not harm past tasks as the fairness weighting is a soft heuristic. This is unlike gradient projection or gradient alignment techniques that explicitly constrain the updates.

Sensitivity to large gradient norms: The fairness mechanism seems to be dependent on gradient magnitudes. It's possible tasks with large gradient norms can dominate the ones with smaller norms. So is the fairness scale dependent?

---

### Official Review · Reviewer_9iqb · 2025-10-31

**Soundness:** 2
**Presentation:** 2
**Contribution:** 2
**Rating:** 4
**Confidence:** 5

**Summary:**

The authors introduce FairOCL, a new way to combine gradients more fairly. They borrow the idea of fairness from communication networks, where multiple users share limited resources. Here, each task is treated like a user, and the gradient direction is the shared resource. FairOCL finds a balance so that no old task dominates or gets ignored. They also add a knowledge distillation trick (using a “teacher” model) to reduce the bias toward recently seen data.

**Strengths:**

1. FairOCL is simple to implement, integrates smoothly with standard replay-based training, and introduces minimal computational overhead compared to existing methods.
2. The paper provides intuitive connection between resource allocation and gradient aggregation in online continual learning. This analogy offers a principled way to think about balancing task influence during replay.

**Weaknesses:**

1. Limited novelty – The idea of reweighting or balancing gradients has been explored before in continual learning. FairOCL mainly repackages this concept under a fairness interpretation.
2. Outdated experimental setup – The evaluation relies mostly on small-scale benchmarks (CIFAR-10/100, TinyImageNet) and older baselines (ER, DER++, DVC). Many stronger recent OCL methods (e.g., DyTox, CODA-Prompt, Co2L, CLS-ER, or MEMO) are missing.
3. Fixed parameter α – The fairness level is tuned manually per dataset; there’s no adaptive or principled selection.
4. Limited realism of fairness evaluation - The paper could better explain how fairness in gradients translates into fairness in learned features. The proposed fairness concept is tested only under balanced class distributions, where every task contributes similarly sized datasets. It would be more meaningful to evaluate FairOCL under imbalanced or long-tailed continual learning settings, where resource sharing constraints are more realistic and the notion of fairness has stronger practical relevance.
5. Missing Literature - Many prior gradient reweighing methods are not cited in this work eg: [1],[2]

citations:
[1] Gradient Reweighting: Towards Imbalanced Class-Incremental Learning (Jiangpeng He, Fengqing Zhu)
[2] Rethinking Gradient Projection Continual Learning: Stability/Plasticity Feature Space Decoupling (Zhen Zhao; Zhizhong Zhang; Xin Tan; Jun Liu; Yanyun Qu; Yuan Xie)

**Questions:**

1. Could you better differentiate FairOCL from prior gradient reweighting methods?
2. How sensitive is performance to α, and could you provide guidance or heuristics for practitioners to choose it? Do different α values correspond to specific stability–plasticity trade-offs that could be described more clearly in the paper?
3. Could you elaborate on how fairness at the gradient level might translate to fairness in learned features or task retention? Is there qualitative evidence (e.g., visualization, theoretical argument) showing that gradient balance leads to more uniform feature preservation?
4. The fairness principle is evaluated on balanced datasets. How would FairOCL behave under imbalanced or long-tailed task distributions?
5. What unique advantage does the fairness-based formulation offer beyond a change in interpretation. For instance, does it provide different gradient solutions or improved theoretical guarantees?

---

### Official Review · Reviewer_itRt · 2025-11-05

**Soundness:** 4
**Presentation:** 4
**Contribution:** 4
**Rating:** 8
**Confidence:** 4

**Summary:**

The authors address the issue of conflicting gradients in online continual learning (OCL). They propose FairOCL, a framework based on fair resource allocation in communication networks. FairOCL formulates gradient aggregation across tasks as a constrained utility maximization problem and enforces fairness in the optimization process, allowing principled control over task prioritization. Proposed technique is novel and the empirical evaluation is exhaustive.

**Strengths:**

1. The idea to frame gradient aggregation in online continual learning as an \alpha‐fair utility maximisation problem inspired by communication networks is a novel and principled approach.
2.  The authors recognise that fair aggregation alone is insufficient due to task recency bias. The inclusion of an EMA‐based teacher and a KL‐divergence loss provides a simple and effective method that improves retention.
3. The authors evaluate FairOCL on four benchmarks with varying buffer sizes and compare against strong replay baselines and state‐of‐the‐art methods. Across all datasets and memory sizes, the proposed method achieves higher average accuracy and lower forgetting. The ablation study demonstrates that FairOCL is robust to the choice of α and distillation temperature.The experiments show the method's stability.

**Weaknesses:**

1. The problem of fairness in continual learning is not well-motivated. How does the problem of gradient conflict translate to fairness? Arguments or specific use cases that demonstrate the presence of gradient conflict and resulting issue of fairness is not provided.
2. This work seems very close in writing to the reference  Wu et al. (2024). Although authors point out the difference clearly, there could be more effort to dealign introduction and other sections.
3. Important tradeoffs may not be analysed. For instance, are there situations where ensuring fairness lead to increase in forgetting or decrease in accuracy. Intuitively, it seems like ensuring fairness tends to have a long term gain and hence, for the first few tasks, I expect the proposed method not to perform well. However, as the task advances (long task sequences), the proposed method may do better and better. However, no such discussions are available and this hinders the intuitive understanding of the work.
4. How does fairness affect stability Vs plasticity ? This needs more discussions.

**Questions:**

1. The introduction states that joint optimisation on current and replayed data produces gradient conflicts, but the paper does not provide examples quantifying how often such conflicts occur, which tasks are adversely affected, or how uniform averaging produces imbalanced retention. Including case studies or synthetic examples illustrating this would strengthen the problem statement and motivate the fairness formulation.
2. Equation (5) introduces a ball constraint parameterised by ε, but ε is neither fixed nor further discussed. It would improve readability to define all variables and symbols and to be consistent in subscripts and superscripts.
3. The authors say that Eq. (7) is solved via a constrained nonlinear least‐squares formulation and that the overall cost remains comparable to other methods. A small table reporting the number of iterations and runtime per batch on each dataset would support the claim that the training budget is comparable.
7. The paper argues that fairness in gradient aggregation yields balanced retention across past tasks, yet no explicit fairness metrics are reported. Plotting per‐task improvements using matrices such as Jain’s fairness index would show if the method is actually fair. Similarly, it would be useful to report task‐wise accuracy at the end of training to verify that earlier tasks do not deteriorate relative to later ones.
8. alpha is a tunable parameter; however this being a hyperparameter becomes an issue. Shiuld this be learned from data?

---

### Note · Authors · 2025-11-13

I have read and agree with the venue's withdrawal policy on behalf of myself and my co-authors.